# Molecular Detection of *Babesia gibsoni* in Cats in China

**DOI:** 10.3390/ani12223066

**Published:** 2022-11-08

**Authors:** Fangyuan Yin, Daoe Mu, Zhuojia Tian, Dong Li, Xiting Ma, Jinming Wang, Guiquan Guan, Hong Yin, Facai Li

**Affiliations:** 1College of Veterinary Medicine, Southwest University, Chongqing 400715, China; yinfangyuan96@126.com (F.Y.); mudaoe@163.com (D.M.); tzj2956@163.com (Z.T.); markleechongqing@163.com (D.L.); mxt15803006845@163.com (X.M.); 2State Key Laboratory of Veterinary Etiological Biology, Lanzhou Veterinary Research Institute, Chinese Academy of Agricultural Science, Lanzhou 730046, China; wjm0403@caas.cn (J.W.); guanguiquan@caas.cn (G.G.)

**Keywords:** *Babesia gibsoni*, *Babesia* spp. in feline, pet cats, 18S rRNA, China

## Abstract

**Simple Summary:**

Tick-borne diseases in companion animals have been increasing globally. Domestic dogs and cats as potential reservoir hosts of tick-borne pathogens might transfer zoonotic diseases to humans. There are currently few reports of feline babesiosis in China. To investigate the incidence of *Babesia* spp. infection in cats, blood samples were collected from Chongqing, Fujian, Hubei and Shandong, and *Babesia gibsoni* was detected. These findings will be useful for understanding the epidemic situation of babesiosis in China and provide a theoretical basis for undertaking effective disease control measures in the interests of public health.

**Abstract:**

As there are few studies of *Babesia* spp. infection in cats in China, or anywhere in the world, the aim of this study was to explore the epidemic features of babesiosis in pet cats in China. In total, 429 blood samples were randomly collected in four different geographical regions. The 18S rRNA gene fragment of *Babesia* spp. was amplified by nest polymerase chain reaction (PCR), and haplotype and phylogenetic analysis of *Babesia* were performed to analyze the relationship of this protozoa. The total positive rate of infection was 2.8%. BLAST analysis indicated that *Babesia gibsoni* was detected in 12 cats. Among these, 4.3%, 3.1%, 0.8% and 2.0% were from Chongqing, Fujian, Hubei and Shandong, respectively. Haplotype and phylogenetic analysis showed that there were nine haplotypes and no obvious genetic variation among *B. gibsoni* populations. These findings will be helpful for understanding the epidemiology of *Babesia* spp. in China, and provide a foundation for developing effective preventative strategies.

## 1. Introduction

Tick-borne diseases are recognized as important infectious diseases, posing a potential threat to the health of humans and animals. Tick-borne infections in companion animals have been increasing worldwide, which could be due to the distribution of vectors influenced by climate change, environmental and artificial factors [1] and increased detection capacity with the popularization of molecular biology. In China, there were 74 million dogs and 67 million cats as companion pets at the end of 2018 [2]. As potential reservoir hosts of tick-borne causative agents, domestic dogs and cats can transfer zoonotic diseases to humans [3,4], so the risk of feline and canine tick-borne pathogens has drawn attention from veterinary and public health research organizations.

Babesiosis, caused by the intracellular hemoprotozoa of the genus *Babesia*, is transmitted by ixodid ticks and has been widely described [5,6]. The severity of this disease ranges from mild infections to severe illness characterized by anorexia, fever, icterus, pallor, splenomegaly, hemolytic anemia and hemoglobinuria [7,8,9]. The species of *Babesia* are divided into the large types, including *Babesia canis*, *Babesia rossi* and *Babesia vogeli* and small types, including *Babesia conradae*, *Babesia gibsoni* and *Babesia vulpes* [10,11,12,13]. In China, *B*. *canis*, *B. gibsoni* and *B*. *vogeli* have been described in dogs and are endemic in the central, southern and eastern regions [14,15,16,17,18]. Feline babesiosis has been reported sporadically in Europe and Asia, whereas it was found to be more common in South Africa [19]. Several *Babesia* species have been detected in domestic and stray cats, including *B*. *canis*, *B*. *cati, B*. *felis, B*. *gibsoni*, *B*. *hongkongensis*, *B*. *lengau*, *B*. *leo, B*. *microti*, *B*. *presentii*, *B*. *vogeli* and unidentified *Babesia* spp. [20,21,22]. In China, *B. vogeli* has been reported in cats in Shenzhen and *B. hongkongensis* was found in cats in Hong Kong and Hunan [22,23,24].

Information on the prevalence and distribution of feline babesiosis in China remains limited. Therefore, the aim of this study was to investigate the incidence of *Babesia* spp. in blood samples in domestic cats from southwestern, central and southeastern regions of China, and describe its genetic relationship according to the regions of collection.

## 2. Materials and Methods

### 2.1. Sample Collection and DNA Extraction

A total of 429 blood samples were randomly collected from domestic cats in pet clinics. These samples were from five cities, including 188 from Chongqing, 51 from Linyi, 64 from Quanzhou, 25 from Wuhan and 101 from Xiangyang, as seen in Figure 1. The age range of the cats were from two months to eight years. Samples were obtained from cats with no obvious clinical signs of *Babesia* infection and no ticks were found on the bodies of these pet cats. Peripheral blood samples were collected in EDTA vacutainers and genomic DNA samples were extracted from 250 μl thawed blood using the Blood DNA Mini Kit (Omega, Norcross, GA, USA) according to the manufacturer’s instructions. DNA samples were stored at −20 °C until use.

### 2.2. PCR Amplification and Sequencing

To identify the *Babesia* infection in cats, a nested PCR was employed to amplify a region of the 18S rRNA gene as previously described [16,18,25]. Primary amplification was conducted using Piro1-S: 5′-CTTGACGGTAGGGTATTGGC-3′, Piro3-AS: 5′-CCTTCCTTTAAGTGATAAGGTTCAC-3′ to amplify a gene fragment of 1379 bp, and secondary amplification was performed using Piro-A: 5′-ATTACCCAATMCBGACACVGKG-3′ and Piro-B: 5′-TTAAATACGAATGCCCCCAAC-3′ to amplify a gene fragment of 405 bp. The PCR reaction conditions were described previously [25]. All PCR products were analyzed on 1.5% agarose gels stained with GoldView II dyes (Solarbio, Beijing, China) and visualized under ultraviolet light. A sick dog was clinically diagnosed as babesiosis-positive by microscopic analysis and a positive PCR test for *B. gibsoni*. Genomic DNA extracted from the blood sample of the sick dog was used as a positive control and distilled water was used as negative control.

The positive amplicons were purified using a Hipure Gel Pure DNA Mini Kit (Magen, Guangzhou, China). The purified products were cloned into the pMD19-T vector (TaKaRa, Dalian, China), and then transformed into *Escherichia coli* Trans5α competent cells (TransGen, Beijing, China). Three positive clones were sequenced using universal M13 forward and reverse primers (PRISM3730XL, ABI).

### 2.3. Sequences Analysis

The obtained 18S rRNA sequences were analyzed using the NCBI BLASTN program (https://blast.ncbi.nlm.nih.gov, accessed on 27 January 2022 and 22 June 2022) and the sequences were deposited in GenBank under the accession numbers OM403679-OM403682 and ON810481-ON810488.

Multiple sequences were aligned using Clustal W within MEGA 11 software [26]. Nucleotide sequence analysis was performed by Genedoc program [27]. A haplotype network was drawn using the TCS network within PopArt software [28,29]. To evaluate the phylogenetic relationships, the sequences were compared with the registered sequences from different countries in the GenBank database. A phylogenetic tree was constructed using the neighbor-joining method based on the Tamura 3-parameter substitution model with gamma distributed (G) rates in MEGA 11, with bootstrap values of 1000 replicates [26]. A 50% cut-off value was performed for the consensus tree.

## 3. Results

### 3.1. Detection and Identification of Babesia spp.

The results showed that the total prevalence of *Babesia* spp. infection was 2.8% (12/429) of sample cats, shown in Table 1. Among these 12 positive samples, 4.3%, 3.1%, 4.0% and 2.0% were from Chongqing, Quanzhou, Wuhan and Linyi cities, respectively, but there were no positive samples found in Xiangyang city. All the positive pet cats were under one year of age. Blast analysis showed that the obtained sequences were 98.8% to 99.8% identical to that of *B. gibsoni* from dogs in China (KP666166), India (MN134517), Japan (AB478328) and the USA (DQ184507). Compared with the sequences from China, India, Japan and the USA, these results showed 11 substitutions at nucleotide sites 11, 13, 19, 83, 109, 163, 198, 250, 285, 289 and 360, as seen in Figure 2. All obtained 18S rRNA sequences showed 98.5% to 100% nucleotide identity with each other. Haplotype analysis indicated that nine haplotypes existed in *B. gibsoni* isolates in this study, as shown in Figure 3.

### 3.2. Phylogenetic Analysis of B. gibsoni Using 18S rRNA Sequences

A neighbor-joining (NJ) tree was constructed using the 18S rRNA sequences of 12 *B. gibsoni* isolates derived from cats, together with the data deposited in GenBank (see Figure 4). The results showed that no obvious sub-clusters were observed among *B. gibsoni* isolates from this study and those from other geographic regions including China, India, Japan and the USA. The NJ tree also indicated that there was no apparent genetic variation among the *B. gibsoni* isolates between dogs and cats. The other *Babesia* species such as *B. felis*, *B. hongkongensis*, *B. leo*, *B. microti* and *B. vogeli* formed separate clades with high bootstrap support (Figure 4).

## 4. Discussion

For feline *Babesia* species, *B. vogeli* was found in three of 203 cats in China [22], and has been reported in Thailand, Portugal and Brazil [30,31,32]. In other studies, one cat was diagnosed with *B. hongkongensis* in Hong Kong and two cats were positive for *B. hongkongensis* in Hunan [23,24]. For the presence of *B. gibsoni* in cats, there were related reports in China, Singapore and St Kitts [21,33]. *B. gibsoni* was considered to be a species responsible for canine babesiosis in China, including Shanghai, Jiangsu, Shandong, Anhui, Zhejiang, Jiangxi, Fujian, Hubei and Shaanxi with positivity 0.72% to 64.2% [15,34,35]. 

In the present study, a small-scale survey was conducted in pet animals to investigate the epidemiology of babesiosis. BLAST analysis showed that the obtained 18S rRNA sequences shared high identity with the 18S rRNA of *B. gibsoni*, indicating that only *B. gibsoni* was found in cats and the total prevalence of *B. gibsoni* infection was 2.8%, indicating that the prevalence of *B. gibsoni* infection in cats was lower than in dogs in China. According to clinical records, the pet cats spent most of their time indoors and had limited chance to roam around outside, so were exposed to a low-risk environment with regard to active *Haemaphysalis longicornis* and *Rhipicephalus sanguineus*. All 12 positive cats were under one year of age, which was consistent with a previous study indicating young animals were susceptible to *B. gibsoni* infection [22].

Nucleotide sequence analysis suggested that nucleotide variations were found within 18S rRNA sequences of *B. gibsoni* (Figure 2). Haplotype analysis demonstrated that genetic differences were observed among *B. gibsoni* sequences (Figure 3). Phylogenetic analysis displayed that all *B. gibsoni* 18S rRNA genes from cats belonged to a clade consisting of those from dogs in other parts of China, India, Japan and the USA (Figure 4). These data were in agreement with previous studies, showing a limited genetic relationship of *B. gibsoni* populations in Asia and the USA [36]. *B. gibsoni* has been detected in dogs from Fujian, Shandong and Hubei in the previous studies [15,34]. In this study, *B. gibsoni* was also found in cats from the same three regions. *H. longicornis* occurred in Hubei and Shandong and *R. sanguineus* was endemic in Fujian and Shandong [37,38]. Therefore, it is hypothesized that *B. gibsoni* might be circulating in dogs and cats in these three sampling sites. These results provided evidence of the occurrence of cross-species transmission in the different hosts, which could be related to the movement of humans carrying pets, the mobility of hosts with ticks and shared habitats between different hosts. No ticks were found on the bodies of these pet cats, and what causes *Babesia* infection in pets will be examined in further research. Possibilities include that the ticks had already dropped off the cats before they were taken to the clinic, and it can be inferred that cats could become positive from direct transmission such as fighting and blood transfusion according to the transmission routes of *B. gibsoni* [39,40].

## 5. Conclusions

*B. gibsoni* was found in a low proportion of asymptomatic cats in China, as nine haplotypes found among 12 isolates. Phylogenetic analysis indicated that there was no obvious genetic variation among *B. gibsoni* populations based on 18S rRNA sequences. These findings can provide greater insight into the distribution of *Babesia* and its genetic relationship in these four regions of China, and will be also useful for making effective control approaches to improve the health and welfare of companion animals. 

## Figures and Tables

**Figure 1 animals-12-03066-f001:**
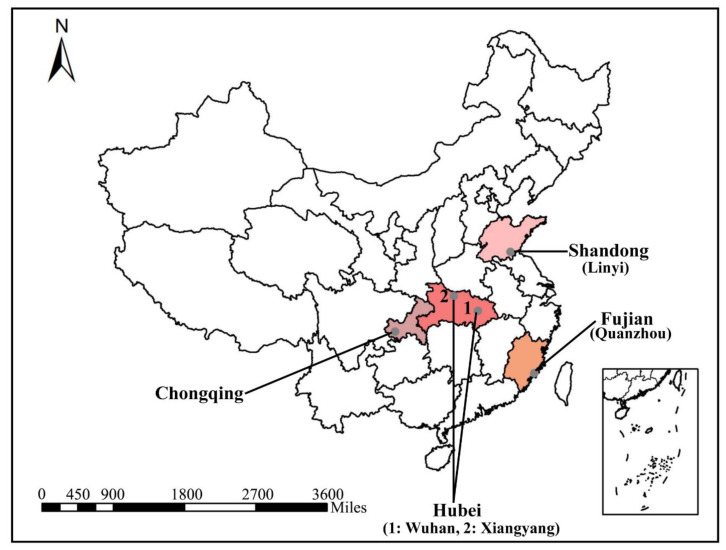
Geographical distribution of sampling sites. Feline blood samples from three provinces (Fujian, Hubei and Shandong) and one municipality (Chongqing) in China.

**Figure 2 animals-12-03066-f002:**
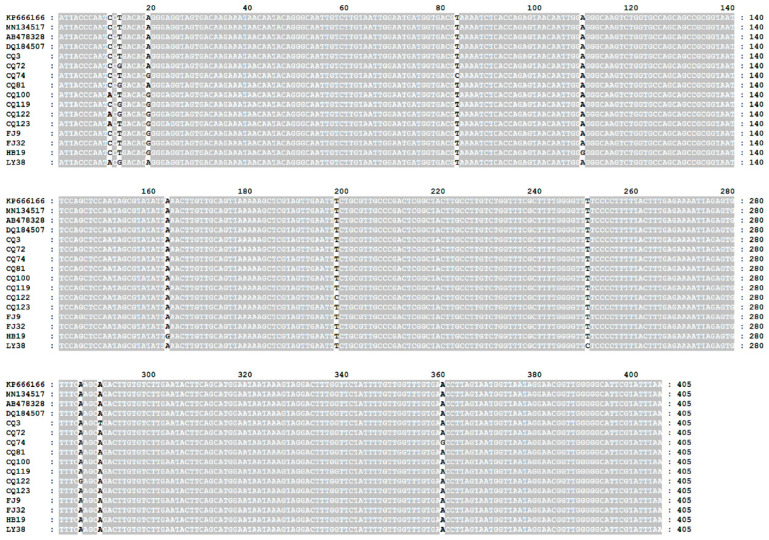
Multiple sequence alignments of *B. gibsoni* in cats in China. The sequences obtained from this study were compared with the related sequences deposited in GenBank, KP666166 (China), MN134517 (India), AB478328 (Japan) and DQ184507 (USA). The nucleotide substitutions occurred at 11 positions including 11, 13, 19, 83, 109, 163, 198, 250, 285, 289 and 360. Abbreviations: CQ (Chongqing), FJ (Fujian), HB (Hubei), LY (Linyi).

**Figure 3 animals-12-03066-f003:**
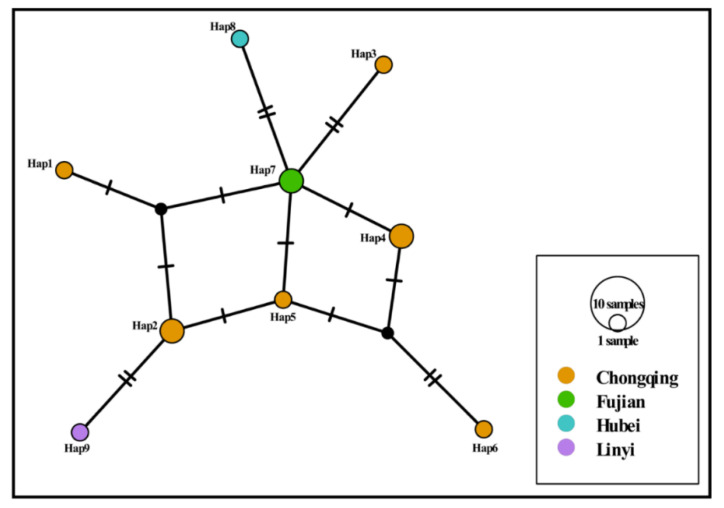
TCS haplotype network of *B. gibsoni* isolates in cats in China. The different coloured dots represent haplotypes from the different locations.

**Figure 4 animals-12-03066-f004:**
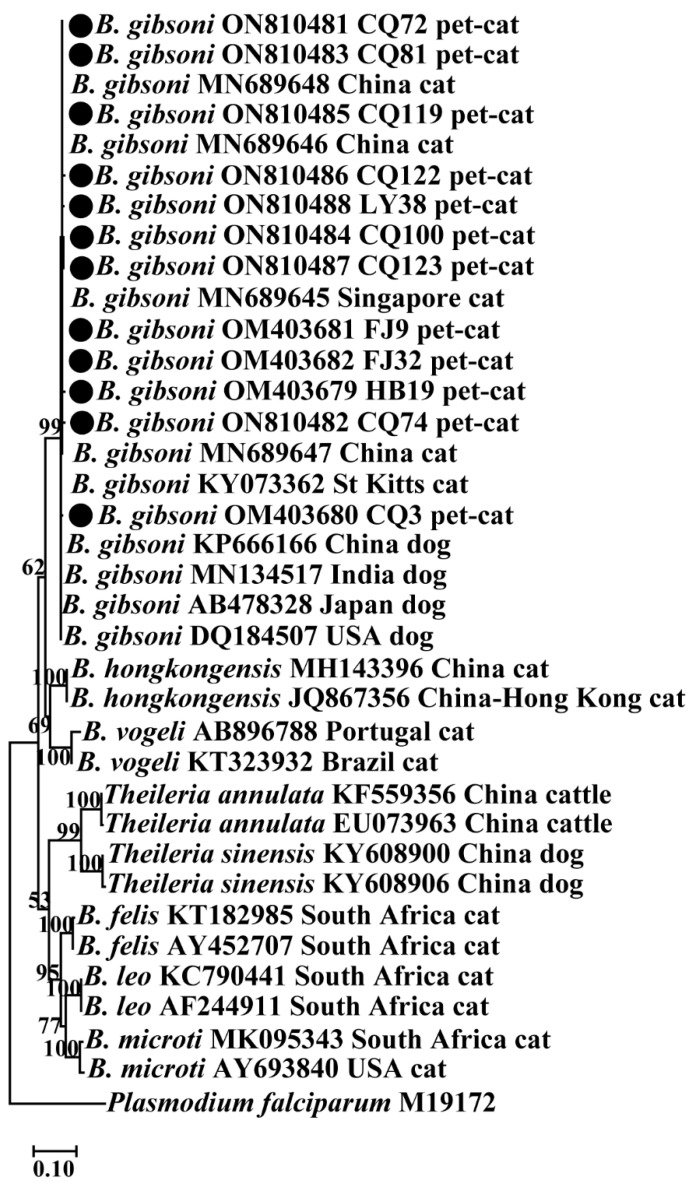
Phylogenetic tree constructed using 18S rRNA sequences from *Babesia* spp. by neighbor-joining. The sequences obtained from this study were compared with the related sequences deposited in GenBank. The bootstrap values of >50% were displayed at each branch point. GenBank accession numbers, the isolate, countries and host were shown alongside species names. The sequence of *Plasmodium falciparum* (M19172) was used as outgroup. Representative isolates in this study were indicated by bold circles. Abbreviations: CQ (Chongqing), FJ (Fujian), HB (Hubei), LY (Linyi).

**Table 1 animals-12-03066-t001:** Prevalence of *Babesia gibsoni* in cats in China.

Locations		No. of Samples	No. of Positive Samples	Positive Rate (%, 95%CI)
Province/Municipality	City			
Chongqing	Chongqing	188	8	4.3 (95%CI: 1.85–8.21%)
Fujian	Quanzhou	64	2	3.1 (95%CI: 0.38–10.84%)
Hubei	Wuhan	25	1	4.0 (95%CI: 0.10–20.35%)
	Xiangyang	101	0	0
Shandong	Linyi	51	1	2.0 (95%CI: 0.05–10.45%)
Total		429	12	2.8 (95%CI: 1.45–4.84%)

## Data Availability

All data generated or analyzed during this study are included in this published article.

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
