# Peer review of "Molecular Detection of Babesia gibsoni in Cats in China"

_animals, 2022, doi:10.3390/ani12223066_

Round 1
Author Response
We thank Reviewer very much for constructive comments and suggestions.
Please see the attachment.

Reviewer 2 Report
The manuscript is the first report of B. gibsoni infection in cats in China. It’s an important finding to understand the epidemiology of B. gibsoni in China. Following comments may help to improve the manuscript before accepted.
Main comments:
1: Accession numbers OM403679-OM403682 and ON810481-ON810488 are not online yet, authors should submit the manuscript to the journal until all sequences online.
2: Universal primers are not enough to present the conclusions, B. gibsoni specific primers are needed to confirm at least for those positive samples.
3: Phylogenetic analysis is poor, a tree based on variable region of TRAP may showing clear genetic variation, not 18S rRNA sequences. That’s the reason why no obvious genetic variation among B. gibsoni populations.
Minor comments:
1: Lines 82-84, “Primary amplification was conducted amplify a gene fragment of 1400 bp”, 1379 bp may correct for B. gibsoni, please confirm.
2: Lines 177-179 needing a reference to support B. gibsoni transmission through fighting and blood transfusion.
Author Response

(The authors gave the same response as above.)

Reviewer 3 Report
Dear authors, your paper is interesting but it should be improved - first, the editing of English is necessary, second material and method regarding animals should be maximally improved and limitations should be included. There are minor points in the text. As I am not an expert on PCR methodology I will not comment. According to me, the title should contain the word prevalence as it sounds like you described one case only.

Author Response

(The authors gave the same response as above.)

Round 2
Reviewer 2 Report
All comments were improved, I think the persent MS is ready for accepted.